# Distorted Body Image and Mental Pain in Anorexia Nervosa

**DOI:** 10.3390/ijerph20010718

**Published:** 2022-12-30

**Authors:** Natalia Ciwoniuk, Magdalena Wayda-Zalewska, Katarzyna Kucharska

**Affiliations:** Institute of Psychology, Cardinal Stefan Wyszynski University in Warsaw, 01-938 Warsaw, Poland

**Keywords:** body image, anorexia nervosa, mental pain, eating disorders, depression

## Abstract

(1) Background: Body image is being defined as the picture of our own body in our mind with its size and shape, and with a perceptive and attitudinal evaluation of this body. It appears to be a complex clinical construct predisposing an individual to developing and maintaining anorexia nervosa (AN), as well as having considerable impact on prolonging the duration of this illness and its relapse risk. The aim of the research work was to assess whether the symptomatology of eating disorders, level of depression, and mental pain are associated with body image, and examine the influence of a distorted body image as well as eating disorders and depression symptoms on mental pain in AN; (2) Methods: A total of 36 women diagnosed with AN and 69 healthy controls (HC) participated in this study. All participants completed a battery of the following scales: EAT-26, BSQ-34, BIDQ, BDD-YBOCS, CESDR, and the Mental Pain Scale; (3) Results: Results show statistically significantly greater body image disturbances and higher level of depression and mental pain intensity in the AN compared to the HC group. Regression analysis indicates a greater impact of distorted body image, eating disorders, and depression symptoms on mental pain in AN; (4) Conclusions: As assumed, distorted body image and mental pain are central components of AN that should be especially emphasized in the therapeutical process of treating AN. Future research should focus on the etiopathogenesis of distorted body image in relation to the chronicity of mental pain and depression in AN, and address these outcomes in clinical practice to minimize suicide risk in this high-risk group of patients.

## 1. Introduction

Body image has been conceptualized as a complex clinical construct including, (a) body schema and (b) the emotional relationship that one has with one’s own body. According to this framework, body schema is defined as a cognitive-informative aspect of the body. It relates to the overall “body of knowledge” that one has about their body (independent of their emotional involvement). Emotional relationship, in contrast to the body schema, consists of a behavioral, emotional, and cognitive component, and refers to the level of satisfaction with one’s own body and the emotional responses that this level of satisfaction elicits [1]. The concept of perceptive-attitudinal evaluations of one’s body [2] points to two components of body image, namely perception—described as the evaluation of bodily accuracy, and attitude—described as a global subjective satisfaction with or affect towards the body. From this point, body image is determined with appropriate proportion between ideal and actual body shape and the greater the converge between these two is achieved, the greater the level of body image satisfaction [3].

Body image constructs as a core symptom of anorexia nervosa (AN) was used for the first time in a study published in 1962 [4]. The DSM-5 manual [5] identifies the following BID criteria that apply to the diagnosis of AN: diagnostic criteria B (“intense fear of gaining weight or becoming obese or persistent behavior to prevent weight gain, even despite considerably low body mass”) and C (“disturbance in the way in which one’s body weight or shape is experienced, undue influence of body weight or shape on self-esteem, or persistent lack of recognition of the seriousness of the current low body weight”) [6]. Patients suffering from AN tend to overestimate the size of their body, particularly their ‘aversive’ body parts such as hips, buttocks, tummy, and arms [7]. Disordered perception of their own body leads to increased anxiety, body disapproval, feeling guilty, and social withdrawal related to their appearance. Some authors [8] emphasize the role of the higher level of discrepancy between the evaluation of the real self and ideal self in AN. The change in this distance is the best indicator of improvement or worsening of the course of anorexia nervosa [9].

The Body Dysmorphic Disorder (BDD) construct has existed in psychiatric literature since the last century and its essence is a preoccupation or excessive concern in a normal-appearing person [10]. In AN, BDD is one of the most difficult in treatment and changing symptom because of patients’ perception and cognitive focus on the imagined defects accompanied with a high level of anxiety and distress [11]. It is important to note that individuals with a diagnosis of AN perceive the body shape or weight of those around them—including other patients—as normal and adequate [12]. 

The subject literature discusses the hypotheses about the occurrence of distorted body image already in the prenatal period [13,14,15], which remains consistent with the neurodevelopmental concept of AN. The psychoanalytic approach to the development of eating disorders, including AN, is based on the belief that attachment to the mother as a primary object in the childhood plays a key role in representing one’s own body in adolescence. Insecure attachment causes frustration, fear, and anger in the child, and returns in the form of symptoms and results with a dependent, weak, and disintegrated Ego (Self) in adolescence [16,17].

Unlike distorted body image, which has been widely described, the concept of mental pain is a relatively new construct in the research on eating disorders [18]. In the subject literature, this term is most often used in the context of self-destructive and suicidal behavior, and is defined as experiencing anguish, suffering, and negative emotions such as terror, despair, fear, grief, shame, guilt, lack of love, loneliness, and loss [19,20]. In the psychodynamic view, mental pain is seen as a feeling of longing and mourning that occurs after the traumatic loss of a loved one [21]. Other researchers propose a so-called integrative model in which mental pain overlaps with depression and partly with feelings of hopelessness. According to this theory, individuals with the highest levels of mental pain are at the highest risk of suicide, as this factor has been shown to be an element significantly indicative of self-destructive tendencies [22,23]. The concept of mental pain proposed by Meerwijk and Weiss [24] refers to feelings that result from perceiving inadequacies of one’s body: patients diagnosed with AN do not accept their bodies, which causes them many difficult and unpleasant emotions and increased social withdrawal or self-aggressive behavior. It is aligned with the concept of AN as a form of rejection of one’s own body in relation to many unpleasant and difficult feelings, analogous to the emotions experienced by those who have tried to take their own lives in the past [25]. AN patients are more than five times more likely to die of any cause compared to healthy age-matched controls, with one in five resulting from suicide [26]. There is a confirmed association between a diagnosis of anorexia and the occurrence of suicidal thoughts as well as the level of depression, which may aggravate the symptoms of AN [27]. Research conducted on a group of 30 patients diagnosed with AN showed that 23% of them tried to commit suicide [25]. Based on the literature on the subject, it can be concluded that the probability of committing suicide is higher in patients with a longer history of the disease and that suicidal tendencies appear more frequently in patients diagnosed with the subtype of purging AN [28]. These conclusions are also supported by the results of other studies [29], which reveal that the reduced quality of life characteristic of patients diagnosed with AN increase the probability of suicidal thoughts. Concluding, identifying the mental pain in AN is the important step to help address outcomes in clinical practice and to minimize suicide risk in this group of patients.

This study aimed to assess BID in women with a diagnosis of AN and to analyze the relationship between BID and the severity of eating disorder symptoms, depression, and perceived mental pain compared to a group of healthy individuals. The assessment of mental pain in AN represents the innovative value of the project, as research on mental pain to date has particularly focused on suicidal or auto-aggressive behavior such as *Non-Suicidal Self-Injury* (NSSI) and its psychopathological conditions such as depression, anxiety, borderline personality disorders, loss, and trauma or child sexual abuse (CSA).

## 2. Materials and Methods

### 2.1. Participants

A total of 105 women aged 18 to 45 years were involved in the study; all participants were of Polish origin. A total of 36 women were patients with a clinical diagnosis of anorexia nervosa, assessed from June 2020 to April 2021. To be enrolled in the clinical group, the clinical diagnosis had to be given by a psychiatrist. To ensure a homogeneous sample, comorbid psychiatric diagnoses of personality disorders, bipolar affective disorder, psychoses, as well as habitual substance abuse (drug or alcohol) were exclusion factors as well as neurological disorders and pregnancy. The healthy control subjects consisted of 69 women who were properly matched to the age of the clinical group. They were recruited based upon clinical measures as well as clinical interviews according to the DSM-5 to exclude the presence of any eating or other psychiatric disorders. Inclusion criteria for the healthy sample were: (1) normal body mass index (BMI) in the range from 18.5 to 24.9 kg/m² and (2) no current diagnosis of any mental disorder according to the DSM-5. 

### 2.2. Procedure

The recruitment process was conducted via invites available at outpatient and inpatient eating disorders settings (with a confirmed clinical diagnosis of AN by a psychiatrist), and both Instagram and Facebook AN groups. At the beginning, participants fulfilled a survey and if they matched the including criteria, they were invited to the next stage of the recruitment process. Subjects read the study description and signed the informed consent sheet prepared in concordance with the current version of the Declaration of Helsinki. They signed the consent for the use of data by the UKSW in accordance with the General Data Protection Regulation. Afterwards, they filled in a battery of psychological measures. There was no remuneration for participating in the study. The study was approved by the local ethics committee (the Institutional Review Board; nr. 01a/2020). 

### 2.3. Measures

Recruited participants (both with diagnoses of AN and HCs) were asked to provide their weight and height to calculate their BMI, which data have been confirmed during the meeting with a researcher, and to complete the following instruments:*(a)* Body Shape Questionnaire (BSQ-34) [30]: belongs to self-report measures and contains 34 test items. Its purpose is to examine a person’s well-being in relation to their appearance over the last four weeks. The higher the average test score, the higher the level of body dissatisfaction (Cronbach’s α = 0.97).*(b)* Yale Brown Obsessive Compulsive Scale modified for Body Dysmorphic Disorder (BDD-YBOCS) [31]: is a tool designed to capture the changes in the severity of obsessive-compulsive symptoms. It is used to study dysmorphic disorders of the body. It consists of 12 closed questions, two of which have been extended with additional qualitative questions (Cronbach’s α = 0.87).*(c)* Body Image Disturbance Questionnaire (BIDQ) [1]: this questionnaire assesses beliefs about physical appearance. The tool consists of 5 open questions and 7 closed questions in which answers are given on the Likert scale from 1 to 5. This tool assesses body image disturbance on a continuum including body image dissatisfaction, distress, and disfunction. BIDQ positively correlates also with body image dysphoria (Cronbach’s α = 0.86).*(d)* Eating Attitudes Test (EAT-26) [32,33]: is a questionnaire for the assessment of eating disorders, enabling the assessment of eating habits and the degree of interest in the weight and size of one’s own body. It consists of two parts: A—contains 26 items and B—contains 6 items. Scores above 20 points indicate harmful eating behavior and a high probability of eating disorders (Cronbach’s α = 0.97).*(e)* Center for Epidemiologic Studies Depression Scale-Revised (CESD-R) [34]: is a self-report scale that examines the occurrence of symptoms of affective disorders, in particular depressive mood, over the past two weeks. The tool contains 20 test items on well-being and behavior. The answers are given on a Likert scale from 0 (not at all or less than 1 day) to 4 (almost every day for 2 weeks) (Cronbach’s α = 0.95).*(f)* The Psychache Scale [20]: is a tool based on Shneidman’s definition of psychological pain, which is pain associated with psychache. It is a measure of self-esteem constructed from 13 test items, assessed on a 5-point Likert scale. This scale’s aim is to check the relationship of psychological pain with the appearance of suicidal thoughts and activities among the population of healthy people and people diagnosed with depressive disorders (Cronbach’s α = 0.98).

### 2.4. Statistical Analysis

#### 2.4.1. Mann–Whitney U

Before conducting the main analyses, we assessed the data for the presence of outliers. We assessed the normality of distribution of the variables with the Shapiro–Wilk test and the following criteria: absolute values for skewness and kurtosis lower than 2.00. Since the distributions of the variables were far from the normal distribution and since the examined groups were not equal, non-parametric tests were used in the study. The analysis of the distorted body image variable was performed using the Mann–Whitney U test for independent samples, and the strength of the association was calculated using Cohen’s r; missing values were excluded case-wise. To reduce the chances of obtaining false-positive results (type I errors), we used the Bonferroni correction.

#### 2.4.2. Correlations

The correlation between distorted body image and the variables of depression and mental pain were then analyzed using Spearman’s rho correlations. 

#### 2.4.3. Regression Analyses

Further, multiple regression analyses were conducted in each of the groups to assess whether distorted body image, eating disorders, and depression symptoms could predict the variance of mental pain. An analysis of standard residuals was carried out, which showed that the data from the AN and the HCs groups contained no outliers (AN: standard residual min = –2.38, standard residual max = 1.89; HCs: standard residual min = –2.44, standard residual max = 2.40). The data also met the assumption of independent errors (Durbin–Watson value = 2.5 in AN and in HC = 1.9). The histogram of standardized residuals indicated that the data contained approx. normally distributed errors, as did the normal P–P plots of the standardized residuals. The scatterplots of the standardized predicted values showed that the data met the assumptions of homogeneity of variance. The data were also checked for multicollinearity (VIF = 1.41–1.63, Tolerance = 0.61–0.71 in the AN group; and VIF = 1.45–1.57, Tolerance = 0.64–0.69 in the HCs group). Some variables were non-normally distributed in both groups. However, the absolute values for skewness and kurtosis were below 2.00; therefore, all values were deemed acceptable and they were entered into the regression models without log-transformation. For the statistical analyses, we used the IBM SPSS Statistics 27 package.

## 3. Results

Table 1 (see below) provides an overview of the AN patients’ and HCs’ characteristics and questionnaire scores. In the AN group, the mean age was M = 24.08 (*SD* = 6.13) and in the HCs group, the mean age was M = 22.90 (*SD* = 2.42); there were no statistically significant differences in age for those two groups (median = 23,000; *p* < 0.350). There was a statistically significant difference in the BMI variable: BMI in the clinical trial was M = 16.76 (*SD* = 1.60) while in the control group, the mean BMI was M = 23.54 (*SD* = 4.64) (median = 20,305; *p* < 0.001). The Mann–Whitney *U* Test was used to test the hypothesis of a significantly higher prevalence of distorted body image in the AN group compared with the HCs (see: Table 1 and Figure 1):

As expected, there were statistically significant differences (*p* < 0.001) between the healthy controls group and the group of women with an AN diagnosis in terms of the dependent variable of body image for all scales used in the study. In addition, the group of women with a diagnosis of AN had significantly higher levels of clinical symptoms such as depression and perceived mental pain compared to the HCs group (the Mann–Whitney *U* Test; *p* < 0.001). The correlations between distorted body image and mental pain, depression, and eating disorders symptoms were also explored in each group (Table 2):

Significant positive correlations were found between distorted body image measured with three independent tools and all variables included in the study in both groups. Based on the analyses obtained, the strongest positive correlation in the AN group was found between body image and mental pain (with highest results in the BIDQ test), and in HCs group the strongest positive correlation was between eating disorders symptoms and body image (BSQ-34).

The results of the regression analyses indicate that the predictor variables (distorted body image, eating disorders, and depression symptoms) significantly increased the amount of explained variance in the mental pain scale scores (see Table 3).

A hierarchical regression analysis was performed to verify whether distorted body image would predict the intensity of mental pain characteristics above and beyond the covariates of eating and depressive symptoms in the AN and HCs groups. The independent variables were selected based on a literature review and the obtained correlation matrices. Results for the AN group in Model 1 indicate that eating disorders and depressive symptoms were statistically significantly associated with mental pain intensity. In Model 2, eating disorders symptoms and depressive symptoms were still statistically significant predictors of mental pain intensity as well as distorted body image. What is important is that the R value increased in Model 2. Overall, the regression equation comprising body image disturbances as the explanatory variable as well as eating disorders and depression as the covariant symptoms showed a positive effect with mental pain as the dependent variable. For the HCs, we found that only depression was statistically significantly associated with mental pain intensity in Model 1. Further, body image and eating disorders symptoms were not statistically significant predictors for mental pain intensity in Model 2. In Model 2, the depression symptoms were statistically significant predictors of mental pain intensity characteristics after partialling out the predictors, body image and eating disorders symptoms.

## 4. Discussion

Based on the analysis of the results obtained in the conducted study, we showed that body image disturbances are statistically significantly more severe in a group of women with a diagnosis of AN compared to a HC group. Moreover, body image disturbances increased mental pain intensity in the AN group but not the HC group. For HCs, only depression was a significant covariant to have mental pain increased, which was confirmed in previous research [22]. This might suggest that the perception of one’s own body and the emotional attitude towards it are two of the pathognomonic symptoms of AN [4,5] and the predictive elements of successful AN therapy [3]. The results obtained are consistent with previous clinical studies [35] in which it has been noted that women with a history of anorexia, after recovery, continue to present greater perceptual distortions and more negative attitudes towards their own bodies compared to a group of healthy participants. Researchers point out poor rates of remission and high levels of relapse [36], and indicate that specialty outpatient treatments do not outperform each other or control comparisons post-therapy or at follow-up [37,38,39,40]. Women with AN experience more suffering and anguish as well as frustration due to dissatisfaction with their own appearance, which affect the depressive and mental pain dimensions, respectively [24,25]. Moreover, they strive to achieve the ideal body shape in their eyes, but are unable to meet their exaggerated demands, which significantly affects their self-esteem [41].

The statistical analyses of the study confirmed the presence of positive correlations between distorted body image and the severity of eating disorders, depression, and mental pain. These results align with other studies reported on AN comorbidity with several other diseases and disorders such as mood disorders, major depressive disorders, anxiety disorders, obsessive-compulsive disorders, developmental disorders among the autism spectrum and attention-deficit hyperactivity disorder, personality disorders, substance abuse, borderline traits, and also schizophrenia [9,10,11,42]. Therefore, in the treatment process of people with a diagnosis of AN, the focus should not only be on the treatment of AN itself, but should also include the treatment of comorbid disorders. Moreover, it might be concluded that the more distorted the body image the patient displays and the more it deviates from her expectations, the greater is the suffering. This corresponds with previous results which show that the perception of body image in AN is strongly and significantly associated with symptoms of depression and anxiety even ten months after the start of therapy, and that the correlation factors related to depression and anxiety reach their highest values at the end of the treatment process [43,44]. This points out the fact that the persistence of a distorted perception of body schema can become increasingly depressing for patients whose weight gradually increases as therapy progresses. Many studies emphasize increased body checking, avoidance, and anxiety following weight restoration [5,45,46]. Accordingly, future research should focus on finding and exploring body image protective factors related to cultural pressures, personality traits, and life experiences [47] since as far as approximately half of the women population are dissatisfied with their weight and overall appearance. There is also a great challenge in improving existing treatment programs for people with eating disorders as well as to construct new, more effective interventions that enable patients to reduce the symptoms that sustain the disease. Failure to work through the conscious and unconscious elements of distorted body image may prevent healing and promote relapse, which in turn prolong the duration of the illness and contribute to the psychological and somatic losses resulting from the illness.

One of the limitations of the study is the relatively small size of the groups—especially the clinical group. The overriding aim in the methodological context was to obtain homogeneous groups based on highly selective recruitment criteria to ensure that the results obtained would be as reliable and factual as possible. Therefore, 15 female volunteers were rejected for participation in the study based on eligibility and exclusion criteria. Another limitation of this study is its cross-sectional design.

Undoubtedly, longitudinal studies conducted again after a longer interval, when the subjects have recovered, could be of greater benefit in research on body image in AN to verify further whether BID remains a state- or trait-dependent clinical construct.

## 5. Conclusions

Concluding, the study shows that body image disturbances are more severe in people with a diagnosis of AN than in healthy women. Furthermore, women with an AN diagnosis present with a higher level of depression and eating disorder symptoms and because of this, experience more mental pain than those without an AN diagnosis. Some research [48] points out greater mental pain related to the greater severity in AN compared with other ED diagnoses such as bulimia nervosa (BN) or binge eating disorders (BED). Mental pain is a mediator of suicidal acts while on the other hand, it mediates the search for change within oneself. In addition, it correlates highly with body image scores and has a strong positive correlation with dysmorphic features. The results of this research confirm the belief that it is a misunderstanding to ignore the phenomenon of mental pain in the process of the diagnosis and treatment of AN patients.

The innovative aspect of the study, however, is the inclusion of the mental pain variable in the analyses of body image, which has not been previously explored empirically in the context of AN.

## Figures and Tables

**Figure 1 ijerph-20-00718-f001:**
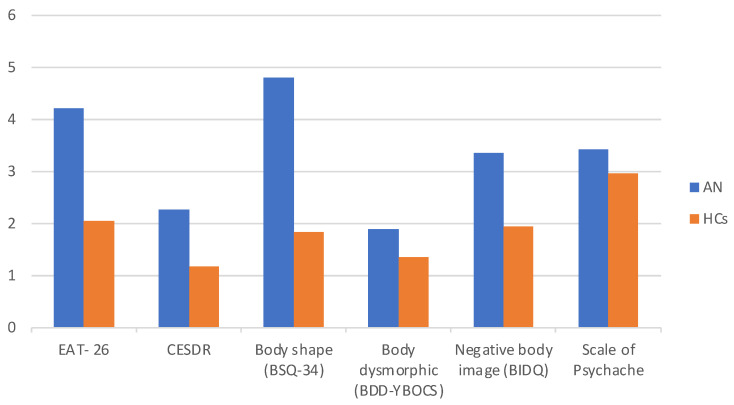
Clinical data for the two subject groups. EAT-26—Eating Attitude Test-26, CESD-R—Center for Epidemiologic Studies Depression, BSQ-34—Body Shape Questionnaire Scale-Revised, BDD-YBOCS—Yale Brown Obsessive Compulsive Scale modified for Body Dysmorphic Disorder, BIDQ—Body Image Disturbance Questionnaire. AN—patients with anorexia nervosa, HCs—healthy controls.

**Table 1 ijerph-20-00718-t001:** Summary of group characteristics and results of the two-way analysis of variance.

	AN	HCs	Mann-Whitney *U* Test		
	*n* = 36	*n* = 69	*U*	*Z*	*p*
	Mean (*SD*)	Mean (*SD*)			
Age	24.08 (6.13)	22.90 (2.42)	1274,5	0.22	<0.825
MI	16.76 (1.60)	23.54 (4.64)	2433.0	8.04	<0.001
EAT-26	4.22 (0.80)	2.05 (0.69)	123.50	−7.56	<0.001
CESDR	2.27 (0.99)	1.17 (0.82)	489.00	−5.09	<0.001
Body shape (BSQ-34)	4.80 (0.90)	1.83 (0.95)	387.00	−5.77	<0.001
Body dysmorphic (BDD-YBOCS)	1.88 (0.37)	1.36 (0.37)	383.00	−5.80	<0.001
Negative body image (BIDQ)	3.35 (0.92)	1.95 (0.68)	312.50	−6.28	<0.001
Scale of Psychache	3.42 (1.03)	2.96 (1.42)	337.00	−6.12	<0.001

Note. BMI—body mass index; EAT—Eating Attitudes Test-26; CESD-R—Center for Epidemiologic Studies Depression Scale-Revised; BSQ-34—Body Shape Questionnaire; BDD-YBOCS—Yale Brown Obsessive Compulsive Scale modified for Body Dysmorphic Disorder; BIDQ—Body Image Disturbance Questionnaire; *U*—Mann–Whitney *U* test.

**Table 2 ijerph-20-00718-t002:** Correlations between indicators of BID and clinical traits in the AN and the HCs group.

		AN			HCs	
	BSQ-34	BDD-YBOCS	BIDQ	BSQ-34	BDD-YBOCS	BIDQ
Scale of Psychache	0.67 **	0.63 **	0.68 **	0.37 **	0.51 **	0.53 **
CESDR Eat-26	0.56 ** 0.63 **	0.51 ** 0.59 **	0.48 ** 0.45 **	0.39 ** 0.78 **	0.57 ** 0.60 **	0.54 ** 0.49 **

Note. ** Correlation is significant at the 0.01 level (both sides).

**Table 3 ijerph-20-00718-t003:** Hierarchical regression analysis predicting mental pain intensity from distorted body image after controlling for the level of eating disorders and depression symptoms.

		B	*SE*	*Β*	*t*	95% CI for B	Adjusted R²	*F*
AN group								
Model 1	Constant	0.35	0.52		0.68	[–0.70, 1.41]	0.62	30.02 ***
	EAT-26	0.47	0.16	0.37	2.94 **	[0.15, 0.80]	ΔR² = 0.65	Δ*F* = 30.02 ***
	CESDR	0.58	0.13	0.54	4.27 ***	[0.38, 1.90]		
Model 2	Constant	–0.17	0.50		–0.33	[–1.20, 0.87]	0.69	26.95 ***
	EAT-26	0.37	0.15	0.29	2.43 *	[0.06, 0.67]	ΔR² = 0.72	Δ*F* = 26.95 ***
	CESDR	0.45	0.13	0.42	3.56 ***	[0.19, 0.70]		
	BIDQ	0.35	0.13	0.32	2.83 **	[0.10, 0.61]		
HCs group								
Model 1	Constant	0.61	0.23		2.61	[0.14, –1.07]	0.55	42.61 ***
	EAT-26	0.14	0.13	0.10	1.12	[–0.11, –0.40]	ΔR² = 0.56	ΔF = 42.61 ***
	CESDR	0.81	0.11	0.70	7.60 ***	[0.60, 1.02]		
Model 2	Constant	0.36	0.27		1.36	[–0.17, 0.89]	0.57	30.45 ***
	EAT-26	0.06	0.13	0.04	0.45	[–0.21, 0.33]	ΔR² = 0.58	ΔF = 30.45 **
	CESDR	0.73	0.11	0.63	6.46 ***	[0.51, 0.96]		
	BIDQ	0.25	0.14	0.18	1.80	[–0.03, 0.52]		

Note. CI = confidence interval; * *p* < 0.05; ** *p* < 0.01; *** *p* < 0.001.

## Data Availability

The datasets generated during and/or analyzed during the current study are available from the corresponding author on reasonable request.

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
