# Peer review of "Distorted Body Image and Mental Pain in Anorexia Nervosa"

_ijerph, 2022, doi:10.3390/ijerph20010718_

Round 1

Reviewer 1 Report

I would like to thank the Editor for the opportunity to review this interesting study and I am flattered to be able to provide my contribution. In general, I find this article to be well written, I do find this paper to be a good discussion issue about distorted body image and mental pain in AN. However, the paper presents some minor weaknesses. I would ask the Authors to address minor amendments, as follows:

a) In the introduction (line 38), DSM-V should be DSM-5.

b) In the section Statistical Analysis, before correlational analysis (line 161), the authors should indicate that for the interpretation of the significant differences between the groups (AN and HC), they used the Bonferroni correction (0.05/8)=0-006. In fact, although they do not indicate, all their differences are at 0.001, so it would be better justified that they don´t increase the type 1 error with the multiple comparisons in relation to the analyses included in Table 1.

Reviewer 2 Report

The work is well planned, statistically it is well resolved, but it suffers from a series of issues that I am going to highlight:

The first sentence of the abstract is like very forceful, and admits a multitude of nuances.

AN, it appears clarified in the abstract but not HC

In sentences like "Regression analysis indicate at impact of distorted body image, eating disorders and depression symptoms on mental pain in AN" it should be clarified what impact and with what meaning or trend

in the conclusions of the abstract; Body image goes from being the determinant of AN to being a component of AN. What has been the process? Then in introduction "body image" is associated with a central symptom. They should be clearer

Introduction and discussion are disconnected. Only reference 5 is common in both sections. Obviously this causes the introduction and discussion to be completely disconnected. A problem arises based on references, which justify an investigation that is later interpreted based on references outside of that initial assessment. I don't want to detract from the work, but it has to be well written and well organized.

I don't see any sense in this statement.

"In this approach, eating disorders pathology is a result of development in insecure relationship between mother and child (attachment) which results with dependent, weak, and disintegrated Ego (Self) in adolescence [16;17]" , also supported by two references one from 63 and another from 73, and I don't see a clear anchorage with the rest of the work.

The concept of "mental pain", they affirm that it is relatively new in research and they initially rely on a reference from 2013, another from 1996, 2017, an idea by Freud, but extracted from a publication from 1955 (which is obviously earlier). ...2009, 2001, 2011 there is one from 2019 although on the relationship between psychological deterioration and suicidal ideation. I do not see well argued that it is something relatively new, nor do I see it demonstrated that it is something of recent interest.

Participants are asked to provide weight and height for the MBI calculation, there is no control over the process? What guarantees are there that the data is correct?

How have the instruments been completed?

The procedure should be more explicit.

Why have you used that number of participants, 36-69, any criteria on the size of the sample?

That the groups are not equal is not a justification for using non-parametric. With such a sample, surely the same results can be obtained using parametric techniques.

The Statistical Analysis section has results, so it should be better organized.

The control group is almost a year younger, although there are no significant differences, it would be important to provide the ranges in all measures. Since, for example, in age, the mean of the AN group is higher, and the SD too, whereas in the control group, the mean is lower but also the SD, so in terms of age it is a group more homogeneous. To continue with another similar example, BMI (with significant differences) shows greater dispersion in the HC group.

Table 1 and Figure 1 show redundant information.

Between lines 192 and 196 they use APA type references, so they would have to review the writing.

What is "EDs characteristics" appears "only" in line 230 and 232 as if by magic. At first I thought of Eating Disorders, but the sentences "Results for AN group in Model 1 indicate that eating disorders and depressive symptoms were statistically significant associates of EDs characteristics. In Model 2, eating disorders symptoms and depressive symptoms were still statistically significant predictors of EDs as well as distorted body image. Since Esting Disorders cannot be prognostic and criterion variables at the same time. In addition, table 3 talks about "mental pain", which is not mentioned in either of those two sentences. I don't quite understand it.

As for the data analysis, it would be interesting for them to carry out a mediation analysis, since the sample size is good, they also obtain correlations between the different variables, so a mediation analysis (SPSS Process Command) could be very revealing in regarding the relationship between the different variables and provides us with more precise information than regression analyses.
